# Pathogenic *E. coli* from Cattle as a Reservoir of Resistance Genes to Various Groups of Antibiotics

**DOI:** 10.3390/antibiotics11030404

**Published:** 2022-03-17

**Authors:** Alexandra Tabaran, Virginie Soulageon, Flore Chirila, Oana Lucia Reget, Marian Mihaiu, Mihai Borzan, Sorin Daniel Dan

**Affiliations:** 1Animal Breeding and Food Safety Department, Faculty of Veterinary Medicine, University of Agricultural Sciences and Veterinary Medicine, Manastur Street No. 3/5, 400372 Cluj-Napoca, Romania; v.soulageon@gmail.com (V.S.); oana.reget@usamvcluj.ro (O.L.R.); marian.mihaiu@usamvcluj.ro (M.M.); mihai.borzan@usamvcluj.ro (M.B.); sorindan@usamvcluj.ro (S.D.D.); 2Microbiology Department, Faculty of Veterinary Medicine, University of Agricultural Sciences and Veterinary Medicine, Manastur Street No. 3/5, 400372 Cluj-Napoca, Romania; flore.chirila@usamvcluj.ro

**Keywords:** bovine, slaughterhouse, *Escherichia coli*, resistance pattern

## Abstract

Antimicrobial resistance has become a worldwide concern in all public health domains and reducing the spread has become a global priority. Pathogenic *E. coli* is responsible for a number of illnesses in humans and outbreaks in the past have been correlated with the consumption of contaminated bovine products. This is why surveillance in all the steps of production is essential. This study focused on identifying the pathogenic strains of *E. coli* in two large bovine abattoirs from Romania and France, and on associating them with the antimicrobial resistance patterns. A total of 250 samples from intestinal content were aseptically collected during the evisceration step of the cattle slaughtering process, from which 242 *E. coli* strains were isolated. Seventeen percent of all samples tested positive to at least one *E. coli* isolate carrying *eae*A, *stx1* and *stx2* genes. The most prevalent genetic profile found in the E. coli strains tested was *Stx1*-positive and *Stx2/eaeA*-negative. More than 68% of the pathogenic E. coli isolated in Romania showed multi-drug resistance (MDR) and in France, the percentage was significantly lower (38%). The MDR profiles showed a high gene diversity for antibiotic resistance, which represents a great risk for environmental spread and human health. Our results indicate that in Romania, bovines can represent a reservoir for MDR E. coli and, hence, a surveillance system for antimicrobials usage in farm animals is highly needed.

## 1. Introduction

In a global context where antibiotics are becoming ineffective, limiting their use becomes a health and environmental emergency [1]. The mechanisms for acquiring resistance are numerous and diverse. Some may be poorly or unrecognized, hence the need to reduce the risk of developing resistance [2]. Reducing the number of resistant bacteria, even commensal bacteria, is a priority [3].

*Escherichia coli* or *E. coli* is mainly a commensal bacterium involved in intestinal function, in the digestion of food and in the supply of certain vitamins. However, *E. coli* can also be pathogenic. There are different types of pathogenic *E. coli*: enterotoxigenic *E. coli* (ETEC), enteropathogenic *E. coli* (EPEC), enteroaggregative *E. coli* (EAgg), enterohemorrhagic *E. coli* (EHEC) and enteroinvasive *E. coli* (EIEC). EHEC strains of *E. coli* are pathogenic for humans and animals. In humans, this bacterium causes food epidemics which result in bloody diarrhoea, haemolytic uremic syndrome (HUS) in children and thrombotic and thrombocytopenic purpura (TTP) in adults [4]. These strains produce toxins called “Shiga toxins”, due to their structural similarities to the toxins of the bacterium Shigella spp., which have the function of damaging the intestinal vascular endothelium of their host [5].

This toxin is encoded by a gene common to all EHEC bacteria, the *stx1* or / and *stx2* gene. In humans, samples and experiments carried out have shown that in order to be highly pathogenic, the *E. coli* must have several concomitant genes [6]. Not only does *stx2* appear to be more pathogenic for humans, but, moreover, the bacterium is only really pathogenic when it has another virulence gene called *eae* encoding an “intimin” protein, making it possible to attach to enterocytes [7]. In cattle, only calves show clinical signs, mainly resembling diarrhoea. As ruminants do not have receptors for *stx*, they have no systemic vascular problems seen in humans and can easily transmit the bacteria through contaminated products [8]. For this reason, cattle are considered to be the main reservoir and the products or by-products obtained from them are considered as a source of infection of *E. coli* producing Shiga toxin (STEC) [9].

STEC have gained increasing global concern worldwide [1,10] and ruminants are regarded as the main animal reservoir [11]. In particular, in geographic regions, such as in some developing countries, the treatment of infections with diarrheagenic *E. coli* are posing a great problem, as this bacterium has managed to become more and more resistant to antimicrobials [12,13]. The wide dispersion of these pathogens in the environment can constitute a serious threat and slaughterhouses are considered an additional source of such contamination with STEC [9].

As previous studies show, cattle are an important source of antimicrobial resistant bacteria (AMR). The resistance in *E. coli* is especially of concern given the risk it poses to human health via the food chain. This fact is also worrying given the possible dissemination of AMR, being in some ways a good indicator of transmission pathways because it is also ubiquitous [9,11]. *E. coli* strains can be found normally in the intestinal tract and bovine carriers are often asymptomatic. Consequently, the transfer of resistance from non-pathogenic strains to pathogenic ones is made easily in the same environment [12]. 

In the 1980s, the first epidemic of haemorrhagic colitis in humans appeared, where *E. coli* serotype O157:H7, originating from minced beef, was involved [14]. In 1995, a toxi-infection from the same strain took place in France and two collective food poisoning outbreaks occurred in 2000 and 2002, for which serotype O157 was involved [15]. In 2003, serotype O157 with H7 antigen was of greatest concern, as the bacterium was found in many species, not only on faecal samples but also directly on carcasses [16]. If we limit ourselves to Europe, in Germany in 1995, many calves suffered from diarrhoea and from those individuals, 122 strains of STEC were isolated. Their genetic study revealed that over 85% of these strains had the *stx1* gene, 10% carried the *stx2* gene, and the rest had both genes [17]. A few years later, in 2002, 4 STEC strains of serotype O26:H11 were discovered in Belgium, possessing several virulence genes, including *stx1*, *stx2* and *eae* [18]. In France, a study shows the presence of *stx* genes in 330 (70%) of 471 faecal samples, of which 34% allowed the isolation of strains of STEC and 9 (5%) had the *eae* gene [19]. The essential point to consider here is that their sensitivity to antibiotics was much greater than that of *E. coli* isolated from cases of bovine pathologies. In 2000, samples from abattoir surfaces revealed the presence of the O157:H7 serotype, some of which carried the *stx1*, *stx2*, *eae* genes [19]. This foreshadows significant contamination of the slaughter line and underscores the importance of bagging the rectum. In Romania, a study conducted on STEC strains isolated over a period of 36 years has shown a concerning increase in multi-drug resistant bacteria [20]. 

This study focused on identifying the *stx* and *eae E. coli* subtypes in two large bovine abattoirs from Romania and France and associate them with the antimicrobial resistance patterns. The objective was to reveal possible variations in resistance patterns given the different surveillance measures in the two countries, and also to underline the importance of this particular food chain in disseminating resistant pathogenic strains. 

## 2. Results

### 2.1. Prevalence of Pathogenic E. coli Strains

Following the isolation protocol, a total number of 242 *E. coli* strains were isolated. All the lactose-fermenting positive colonies developed on MacConjey agar plates were tested by multiplex PCR for the presence of virulence genes encoding Shiga toxins (*stx1* and *stx2*) and intimin (*eaeA*). Seventeen percent (42/242) of all samples tested positive to at least one *E. coli* isolate carrying *eaeA*, *stx1* and *stx2* genes. The positive samples were found in a higher prevalence in the Romanian slaughterhouse investigated (29/42). The highest number may be explained also by a larger number of samples investigated during a longer period of time. The characteristics of *E. coli* in both countries are shown in Table 1.

The most prevalent genetic profile found in the *E. coli* strains tested was *stx1*-positive and *stx2/eaeA*—negative (31/42; 76.1%). None of the strains showed the presence of all virulence genes (*stx1, stx2, eaeA*). Only 11% (5/42) of the positive samples showed the presence of both *stx* genes (*stx1* and *stx2*). One of the strains (2.3%) isolated in France was positive to *eaeA* and *stx2* genes but negative to *stx1*.

### 2.2. Resistance Profiles of Pathogenic E. coli Strains

Figure 1 shows that the highest level of resistance found in all *E. coli* pathogenic strains (*n* = 104) was to nalidixic acid (R—58%; I—4%), followed by tetracycline (R—54%; I—15%) and ciprofloxacin (R—49%; I—6%). The highest level of sensitivity was found to ceftazidime (84%), followed by chloramphenicol (62%). A statistically significant (*p* = 0.004) variation was found in the sensitivity level of the strains from Romania compared to France. While in Romania, most of the samples tested were sensitive to trimethoprim/ sulfamethoxazole (SXT) (79.3%), in France, more than half of the strains tested were found to be resistant (76.9%).

Similarly, a significant (*p* = 0.001) decrease in resistance level was found for cephalosporin class when comparing the strains isolated in the two countries. In this respect, in Romania 31% (9/29) of the samples have shown resistance to one of the antibiotics found in this class (CTX, CAZ) compared to only 10% (1/13) for the strains isolated in France. Additionally, more than 68% (20/29) of the pathogenic *E. coli* isolated in Romania showed multi-drug resistance (MDR). In France, the MDR strains were found in a lower percent (38; 5/13), showing again statistically significant differences (*p* = 0.001). The MDR phenotypes in the pathogenic strains found in both countries are shown in Table 2.

### 2.3. Resistance Genes Presence in Pathogenic E. coli Strains

Of the isolates, 4.76% (2/42) were negative for all the resistant genes tested. The most prevalent genes *tet(A)* and *tet(B)* were identified in more than half of the STEC strains (27/42). The *sul1* and *aadA1* genes were detected also in a high percent in the pathogenic strains investigated (52.3%; 22/42). β-lactam resistance genes were detected in 23.8% (10/42) from *blaTEM* and in 2.3% (1/42;) from *blaSHV*. 

## 3. Discussion

The study focused mainly on culturable pathogenic strains of E. coli, revealing that e antimicrobial resistance patterns show a lot of variations according to the abattoirs or animals from which they were isolated. Although this variation exists, there are similarities regarding the classes of antibiotics to which *E. coli* does seem to have a high prevalence of resistance. Our study has shown that STEC isolated from bovine samples has a markedly high resistance to classical antibiotics, such as nalidixic acid (58%), tetracycline (54%) and ciprofloxacin (49%). These findings are in accordance with other studies performed in Mexico that showed even higher resistance percentages to ciprofloxacin (76%) and tetracycline (69%) [21]. This high prevalence in resistance patterns for tetracycline and nalidixic acid could be explained by their frequent use in prophylaxis and digestive conditions in food animal production farms [22]. The overall prevalence of MDR among the pathogenic *E. coli* found in cattle from Romania is high (68%) compared to France (38%), although the percentage might be influenced also by the difference in the number of samples investigated. The lower prevalence of MDR in France might also be explained by the antimicrobial resistance surveillance system RESAPATH, which provides annual reports and data compilation for the primary bacterial species and general isolates from sick animals from each animal sector [23]. In Romania, even if competent authorities implemented a surveillance system for the detection of pathogens, including AMR, this higher MDR prevalence could be the effect of inappropriate or excessive use of antibiotics for therapeutic and prophylactic treatment. When compared to previous studies, the percentage of MDR found in the Romanian slaughterhouse is relatively high. For example, in Egypt, the MDR was 44.4% [24], in South Africa 13.7% [25], and in Jordan 37.1% [26]. However, the study conducted in Mexico [21] has shown a higher prevalence of MDR then the one revealed by our research (7%). Thus, the common conclusion of all studies is that bovines may represent an important reservoir of antibiotic-resistant bacteria, and this idea is also supported by our research. We have shown that not only is there a high prevalence of MDR, but there is also a possibility for contamination with pathogenic strains of *E. coli* which represent a very important health risk. The MDR patterns found were supported also by the identification of resistance genes in almost all the samples investigated. The most prevalent genes were *tetA tet(A)* and *tet(B)* 64%), followed by *sul1* and *aadA1* genes (52.3%). The *tetA* genes were found in all the pathogenic strains of *E. coli* isolated in France. The large number of resistance genes detected is concerning given the fact that the strains in which they were found are considered pathogenic for humans. Given the fact that they were isolated in the gut, they can subsequently easily become disseminated into the environment.

The virulence factors of STEC strains were also compared to multi-drug resistance patterns. The most prevalent resistance patterns associated with *stx1* gene presence was AMP, TET and NA, while for *stx2* it was AMP, TET, SXT and CIP. All the samples that were positive for *stx2* gene presence were also resistant to tetracycline and ciprofloxacin. Of the total number of strains positive for stx1 gene, a high percentage showed multi-drug resistance (92%; 26/28). Additionally, all the strains positive for *eaeA* gene were multi-drug resistant, the most prevalent pattern being to TET, SMX and SXT. This represents a serious concern given the fact that the *E. coli* strains harbouring *stx* and *eae* genes have been incriminated in human pathogenic enterohemorrhagic *E. coli* strains [27]. The gene *stx2* was often associated with the development of haemorrhagic colitis and haemolytic uremic syndrome in humans [28].

## 4. Materials and Methods

### 4.1. Samples and Isolates

A total of 250 samples from intestinal content were aseptically collected during the evisceration step of the cattle slaughtering process. The abattoir in Romania was visited between a period of two months for sample collection (4 times/month), while the slaughterhouse in France was visited twice. In this respect, 170 samples were collected from the slaughterhouse in Romania, while in France, 80 samples were analysed. The samples taken in Romania were transferred to the analysis laboratory within 3 h after collection in a cool box. The samples collected in France were stored in freezing conditions and shipped to the analysis laboratory within one week. After their arrival, all samples were transferred into separate tubes containing 2 mL Luria nutrient broth (LB) and cultured at 37 °C following the steps stated in the ISO 16654:2001 protocol [29]. Briefly, each sample was inoculated on the MacConkey agar plates (Merck, Germany) and incubated overnight at 37 °C. 

### 4.2. Bacterial Genomic DNA Extraction

DNA extraction followed a protocol previously described by Mihaiu et al. (2014) [30]. Briefly, 2–3 *E. coli* specific colonies were removed from the MacConkey media with a sterile loop and resuspended in 150 μL Chelex solution (Sigma Aldrich, St. Louis, MO, USA). Samples were afterwards subjected to a high temperature protocol (94 °C–15 min and 56 °C–10 min) for cell membrane lysis. DNA quality and quantity for further use was assessed on a Nanodrop ND-1000 spectrophotometer analyser (NanoDrop Technologies, Wilmington, DE, USA).

### 4.3. Virulence Gene Identification

The PCR multiplex method was used for detecting the genes encoding Shiga toxins (*stx1* and *stx2*) and intimin (*eaeA*). The PCR assay was conducted in a final volume of 50 μL comprising the following: 1× PCR green Buffer, 1.5 mM MgCl2; 10 pmol of each primer, deoxynucleotides (dNTPs) each at 200 μM, 1.25 U of Taq DNA polymerase (Promega, Madison, WI, USA), and 5 μL of genomic DNA in a concentration of 50 ng μL^−1^. PCR was performed under the following conditions: 94 °C for 3 min followed by 25 cycles of 94 °C for 30 s, 54 °C for 45 s, and 50 °C for 1 min, and a final extension step of 70 °C for 3 min. *E. coli* strain 152–2 (5) (*eae/stx1/stx2*) was used as positive control and *E. coli* DH5a was the negative control in all tests. Positive and negative controls were previously isolated in the Microbiology Laboratory in the University of Agricultural Sciences and Veterinary Medicine Cluj-Napoca.

### 4.4. Susceptibility Testing

Antimicrobial susceptibility of the pathogenic *E. coli* strains isolated was determined by disc diffusion method as recommended by the Clinical and Laboratory Standards Institute [31] and interpreted as susceptible, intermediate and resistant. Isolates were examined against several classes of antibiotics: Penicillin testing ampicillin (AMP, 10 μg), cephalosporins testing cefotaxime (CTX, 30 μg), ceftazidime (CAZ, 30 μg), macrolide testing chloramphenicol (CHL, 30 μg), quinolones testing ciprofloxacin (CIP, 5 μg) and nalidixic acid (NA, 30 μg), aminoglycoside testing gentamicin (GEN, 10 μg), streptomycin (S, 10 μg), sulphonamide testing sulfamethoxazole (SMX, 300 μg), trimethoprim/ sulfamethoxazole (SXT, 1.25/23.75 μg), and tetracyclines (TET, 30 μg). The interpretation protocol was briefly described by Chirila et al. [20]. All samples that tested resistant to three or more antibiotics were classified as being multi-drug resistant (MDR).

### 4.5. PCR Method for Identification of Resistance Genes

Based on the phenotypic resistance pattern, genes conferring resistance to eight classes of antimicrobials were investigated through the amplification of the following genes: *aadA1 for* trimethoprim, *dfrA1* for streptomycin, *qnrA* for quinolones, *aac* for gentamicin, *sul1* for sulfonamides, *blaSHV*, *blaCMY*, *blaTEM*, *blaCTX* for beta-lactams, *ere(A)* for erythromycin and *tetA*, *tetB* for tetracyclines

The PCR protocol was previously described by Chirila et al. [20]. The amplified product (10 μL) was loaded onto agarose gels (2%). The gels were stained with EvaGreen (JenaBioscience, Jena, Germany) and electrophoresed (90 W) for 40 min. Visualization was performed under UV light with a Gel Doc XR+Imager (Bio-Rad, Hercules, CA, USA). Strains of multi-resistant *E. coli* (O157:K88ac:H19, CAPM 5933) were used as positive controls.

## 5. Conclusions

This study highlights the prevalence of pathogenic *E. coli* in two European countries, but most importantly, the high gene diversity for antibiotic resistance, which represents a great risk for environmental spread and human health. Our results indicate that these particular food-producing animals in Romania are a reservoir for MDR *E. coli* and, hence, an improvement of the current surveillance system for antibiotic usage in farm animals is highly needed.

## Figures and Tables

**Figure 1 antibiotics-11-00404-f001:**
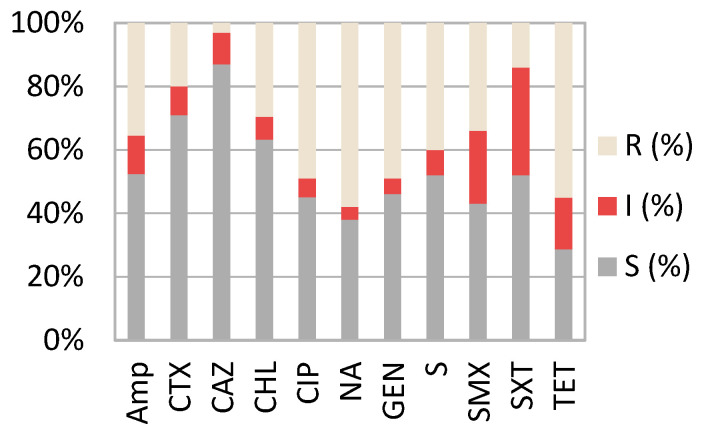
Antimicrobial susceptibility of pathogenic strains. R—resistant; I—intermediate; S—susceptible; AMP—ampicillin, CTX—cefotaxime, CAZ—ceftazidime; CHL—chloramphenicol, CIP—quinolones testing ciprofloxacin; NA—nalidixic acid, GEN—gentamicin, S—streptomycin, SMX—sulfamethoxazole, SXT trimethoprim/sulfamethoxazole, TET—tetracyclines.

**Table 1 antibiotics-11-00404-t001:** Toxin gene presence in positive *E. coli* samples.

Country	No of Samples Tested	Shiga Toxin Genes			Total (%)
		*Stx 1*	*Stx 2*	*eaeA*	
Romania	164	+	−	−	23 (14)
		+	−	+	2 (1)
		−	+	+	0
		−	−	+	1 (1)
		+	+	−	3 (2)
	Total no of samples positive to at least one virulence gene				29 (17)
France	78	+	−	−	8 (10)
		+	−	+	2 (3)
		−	+	+	1 (1)
		−	−	+	0
		+	+	−	2 (3)
	Total no of samples positive to at least one virulence gene				13 (16)

**Table 2 antibiotics-11-00404-t002:** MDR *E. coli* phenotypes and their mechanism of resistance in pathogenic strains isolated in Romania and France.

*E. coli* Strain/Country	Phenotype of Resistance	Resistance Genes	Toxinogenic Genes/Sample No.
1–2/Romania	AMP, TET, SMX, SXT, CTX, NA	*blaTEM, tetA, sul1, dfrA1,* *ere(A)*	*stx1* (1,2)
3–5/Romania 6/France	TET, SMX, SXT, CIP, NA, AMP	*blaTEM, tetA, tetB, sul1, dfrA1,* *ere(A)*	*stx1* (4,6) *stx1, eaeA* (3) *stx1, Stx2* (5)
7/Romania 8/France	AMP, TET, SMX, SXT, NA, CHL	*blaTEM, tetA, tetB, sul1, dfrA1*	*stx1* (7,8)
9/Romania	TET, SMX, SXT, NA, CAZ, S	*blaTEM, blaSHV, blaCMY*, *tetA, tetB, aadA1, dfrA1*	*stx1, eaeA*
10,11/Romania 12/France	TET, SMX, SXT, CTX, S	*tet A, tetB, sul1,* *ere(A)*	*stx1* (10,11) *stx1, eaeA* (12)
13/Romania 14/France	AMP, CIP, TET, GEN	*tetA, tetB,* *ere(A)*	*stx1, Stx2* (13) *stx2, eaeA* (14)
15–17/Romania 18/France	TET, NA, AMP, GEN	*tetA,* *ere(A)*	*stx1, stx2,* (16,17,18) *stx* 1 (15)
19/Romania	TET, NA, GEN, CTX, CIP	*tetA, tetB*	*stx1, stx2*
20/Romania	TET, NA, S,	*tetA, aadA1*	*stx1*
21/Romania	TET, NA, CAZ, SMX, AMP	*tetA, sul1,* *ere(A)*	*stx1*
22/Romania	NA, SMX, SXT, CTX	*sul1*	*stx1*
23/Romania	NA, SMX, SXT, CIP, AMP	*tetA, sul1,* *ere(A)*	*stx1*
24–25/Romania	TET, NA, GEN, CIP, S, AMP	*tetA, sul1, aadA1,* *ere(A)*	*stx1*

AMP—ampicillin, CTX—cefotaxime, CHL—chloramphenicol, CIP—quinolones testing ciprofloxacin; NA—nalidixic acid, GEN—gentamicin, S—streptomycin, SMX—sulfamethoxazole, SXT trimethoprim/ sulfamethoxazole, TET—tetracyclines; *aadA1*—trimethoprim, *dfrA1*—streptomycin, *qnrA*—quinolones, *aac*—gentamicin, *sul1*—sulphonamides, *blaSHV*, *blaCMY*, *blaTEM*, *blaCTX* beta-lactams, *ere(A)*—erythromycin, *tetA*, *tetB*—tetracyclines.

## Data Availability

Not applicable.

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
