# Peer review of "Pathogenic E. coli from Cattle as a Reservoir of Resistance Genes to Various Groups of Antibiotics"

_antibiotics, 2022, doi:10.3390/antibiotics11030404_

Round 1

Reviewer 1 Report

In the presented manuscript, antimicrobial resistance patterns of pathogenic bacteria isolated in bovine slaughterhouses were analysed. Antimicrobial resistance (AMR) is substantial and aggravated problem. This paper is well designed and presented and results are important for those interested in the area of AMR environmental issues. Nevertheless, the manuscript contains some shortcomings. I recommended that this paper should be accepted for publication in your journal after minor revision.

1. Materials and methods section: Why did the Authors choose only one slaughterhouse in Romania and one in France. In my opinion, it is insufficient for broad conclusions. Please, justify it. I suggest completing studies by analyzing more slaughterhouses in these countries in the future. It will be very interesting. 
2. Materials and methods section: Please specify the name of the nutrient broth (line 197)
3. Bacterial genomic DNA extraction section: Line 205: Did The Authors test only the DNA quality (protein contamination etc.)? I suspect that also quantity was tested. Please specify.
4. Discussion section: Authors wrote: "The antimicrobial resistance patterns of E. coli bacteria vary a lot depending on the
countries" - this is not an appropriate statement because only one slaughterhouse in each country was tested. Please correct it.
5. Authors tested only the culturable in lab conditions strains of pathogenic E. coli. Please emphasize this statement in the discussion and conclusions. In my opinion, metagenomic research will be also very interesting, by isolating the DNA directly from the environment. Maybe it's a good direction for future research?
6.  I highly recommend that authors should use language editing services to improve the paper.

Author Response

Dear Reviewer,

Thank you very much for your kind appreciation of our research and your pertinent suggestions for its improvement. In response to your evaluation, we have made the following modifications:

  1. Materials and methods section: In response to your question why we did not study a larger number of slaughterhouses - Unfortunately we had a problem with the owners accepting us in taking samples. We will make sure that our studies continue in the future with more units.
  2. Materials and methods section: (line 197) The name of the nutrient broth was added
  3. Because we used the Nanodrop evaluation we obtained also the quantities. We added the information also in the text
  4. Discussion section: "The antimicrobial resistance patterns of E. coli bacteria vary a lot depending on the countries" - We have corrected the sentence.
  5. Information was added in the first sentence of discussion section (line 157). Our goal is to eventually perform a metagenomic research but for that we have to acquire more samples from the abattoir environments and also farms of origin.
  6. We have improved language editing of the paper asking the help of a native English speaker.

Reviewer 2 Report

The manuscript by Tabaran et al. is original, relatively well designed and conducted. The data provided by the article is an alarm signal regarding the rate of multidrug resistance of E. coli strains isolated from food-producing animals, particularly in Romania, and the risk for environmental spread of MDR strains, as a consequence. I encourage its publication after several modifications as outlined below.

  1. Please correct coli instead E. Coli (line 41)
  2. I recommend to keep the same way of separating the authors cited in the text (line 62 insert , instead ;)
  3. After presenting the abbreviated name of the species Escherichia coli, I suggest the continued use of this form ( coli) throughout the text of the article (38-41 lines).
  4. For genes encoding toxins (Stx1, Stx2) keep the same style throughout the article.
  5. In the chapter Materials and methods please keep the same identification of the subchapters as in the Results chapter (eg 4.1; 4.2 etc.)
  6. Please specify how many series of samples from the Romanian slaughterhouse were collected.
  7. Please provide a brief description of the isolating protocol for coli strains from slaughterhouse samples.
  8. In the Results chapter it is not clear how did the authors selected the coli strains for further testing. The authors need to provide a clear explanation.
  9. Please also specify the percentage of coli strains Stx1-positive and Stx2 / eaeA – negative (line 102), and also of E. coli positive to eaeA and Stx2 genes but negative to Stx1 (line 103-104).
  10. Please insert a paragraph in the text between line 107 and line 113 and also check the font size, which is different from the rest of the text.
  11. I suggest to express as a percentage the susceptibility to trimethoprim/ sulfamethoxazole of the isolated coli strains (lines 112-113).
  12. In text (line 128) table 2 is written with lower case, please correct it (Table 2 instead table 2).
  13. The citation of some authors in the reference list is incorrect (eg. Amézquita-López, B.A., Soto-Beltrán M., Lee B.G., Yambao J.C., Quiñones, B.  instead, Bianca, A.; Amézquita-López, M.S.B; Bertram, G.L.; Jaszemyn, C.Y. I – References no. 11; Imre, K.; Herman, V.; Morar A. instead Kálmán, I.; Herman, V.; Morar, A. - – References no. 13). Please correct these errors.
  14. Lines 260, 265, 267, 2672, 269, 275, 278, 284, 286, 288, 290, 292, 295, 298, 301, 309, 311, 314, 315, 317, 319 the name of the microorganism species are not italicized. Please be carefully with this basic concern throughout the manuscript !!!
  15. I also suggests, the correction of the overall English and scientific style of the manuscript.

Author Response

Dear Reviewer,

Thank you very much for your kind appreciation of our research and your pertinent suggestions for its improvement. In response to your evaluation, we have made the following modifications:

  1. We have corrected coli instead E. Coli (line 41)
  2. We have corrected the text (line 62 we insertedinstead ;)
  3. We have corrected the use of the form ( coli) throughout the text of the article (38-41 lines).
  4. For genes encoding toxins (Stx1, Stx2) we kept the same style throughout the article.
  5. In the chapter Materials and methods we have identified also the subchapters suggested  (eg 4.1; 4.2 etc.)
  6. We added how many series of samples from the Romanian slaughterhouse were collected, in the Material and methods sections
  7. We have added a brief description of the isolating protocol for coli strains from slaughterhouse samples.
  8. We have explained how we have chosen the coli strains for further testing.
  9. We have specified the percentage of coli strains Stx1-positive and Stx2 / eaeA – negative (line 102), and also of E. coli positive to eaeA and Stx2 genes but negative to Stx1 (line 103-104).
  10. We have inserted a paragraph in the text between line 107 and line 113 and also modified the font size to be in conformity with the text.
  11. I expressed in percentage the susceptibility to trimethoprim/ sulfamethoxazole of the isolated coli strains (lines 112-113).
  12. In text (line 128) table 2 we corrected it (Table 2).
  13. The citations of the authors was corrected
  14. All  the names of the microorganism species were italicized.
  15. .We carefully reviewed once again the manuscript and asked the help of a native speaking person.

Reviewer 3 Report

The manuscript “Antimicrobial resistance patterns of pathogenic E. coli isolated in bovine slaughterhouses: comparative study in France and Romania” shows that cattle is  reservoir of E. coli producing Shiga toxin (STEC). The Authors compared the presence of pathogenic E. coli strains in bovine in two abattoirs from Romania and France. In pathogenic E. coli strains was showed the presence  of the virulence genes encoding Shiga toxins (stx1 and stx2) and/or intimin (eaeA). In my opinion it is very important. The Authors also confirmed that cattle represent an important reservoir of resistant bacteria because the pathogenic E. coli strains exhibited resistance to many groups antibiotics. The resistance genes may be transfer rapidly among a number of bacterial species by the horizontal transfer of genes which is a mechanism for the dissemination of multi-drug resistance (MDR) because resistance genes may be in clusters and transferred together to the recipient.

This manuscript can be published in Antibiotics but minor revision is necessary.

The results obtained in this manuscript present  information about resistance of EHCE strains, which EHEC-inducing diarrhea, characterized by bloody stools. Antibiotics should not be used in these cases because they may increase the risk of haemolytic uremic syndrome.

Therefore, the manuscript title should be changed to e.g. “Pathogenic E. coli from cattle as a reservoir of resistance genes to various groups of antibiotics”

Introduction - add information that cattle is an important reservoir of multi-drug resistant microorganisms, and resistance genes can be transferred to other pathogenic species of bacteria

Line  41- instead “E. Coli” should be “E. coli”

Line 45 - instead “spp” should be “spp.”

Gene names should be written in lowercase and italic - please correct it in the manuscript

Table 2 - please explain the designations of the resistance genes

Line 139 - what is beta-Lactamic genes?

Line 169 – instead “isolation” should be “identification”

Line 172 – instead “isolated” should be “detected”

Figure 1 - please add a legend

Line 235 – 238 - please correct the information provided about the investigated genes.

Line 323 - antibiotic susceptibility testing should be based on the latest standards, not the 2010 standards.

Author Response

Dear Reviewer,
Thank you very much for your kind appreciation of our research and your pertinent suggestions for its 
improvement. In response to your evaluation, we have made the following modifications:
The manuscript title was changed as suggested to “Pathogenic E. coli from cattle as a reservoir of 
resistance genes to various groups of antibiotics”
Introduction – we have added the information suggested in a documented paragraph (line 64-71)
Line 41- We have changed to “E. coli”
Line 45 – We have made the modification to “spp.”
All the Gene names have been modified in lowercase and italic throughout the entire manuscript
Table 2 – We have added a footnote to explain the designations of the resistance genes
Line 139 - The phrase was retyped. We were discussing about the resistance genes for B-lactam class of 
antibiotics.
Line 169 – The “isolation” term was modified with “identification”
Line 172 – We replaced the “isolated” term with “detected”
Figure 1 – we added a legend to the figure
Line 235 – 238 - the information provided about the investigated genes was corrected
Line 323 – we updated the protocol to the newest CLSI standard (2020
